# Essential oil-derived decomposable polymers via cycloaddition polymerization of silyl ether-linked phenylpropanoids

Ryo Nagaya[1,5], Tatsuya Seko[1,5], Kazuhiro Okamoto [2], Kazuhide Ueno [1,3], Mahito Atobe[1,3] ✉ & Naoki Shida [1,3,4] ✉

Owing to increasing concerns regarding climate change, research concerning the use of plant biomass as a renewable carbon resource has become increasingly active. Phenylpropanoids, which are aromatic compounds derived from plants, offer renewable sources owing to their availabilities and structural diversity. This study presents an approach for use in producing decomposable polymers with high biomass contents via [2 + 2] cycloaddition polymerization. Bifunctional monomers with silyl ether-linked phenylpropanoids were synthesized, and their polymerizations were investigated using chemical and electro- and photochemical methods. The resulting polymers contained aromatic and cyclobutane rings and silyl ether bonds in their backbones, which enhanced their thermal properties. Notably, these polymers could be decomposed via Diels-Alder reactions at the cyclobutane rings or Si−O bond cleavage, facilitating chemical re- and upcycling. Here, we show a sustainable method of producing high-biomass decomposable polymers, potentially contributing in reducing plastic waste and promoting a circular economy.

The efficient use of renewable plant biomass resources in polymeric materials is urgently required to halt anthropogenic carbon emissions from petroleum-derived raw materials[1–5]. Biomass resources derived from wood and plants display potential as sufficient carbon sources to replace raw petroleum materials because of their levels of abundance. Phenylpropanoids are natural aromatic organic compounds extracted from plants, such as anise, ylang-ylang, and clove, as essential oils. Because of their availabilities and structural diversity, they are widely used in spices, cosmetics, and pharmaceutical derivatives[6–9], and the worldwide market for essential oils reached USD 10 billion in 2021[10]. In addition, considerable efforts have been devoted to generating phenylpropanoids directly or indirectly from inedible woody biomass, such as lignin, facilitating its use as a renewable source of raw materials[11–14].

In this context, the polymerization of phenylpropanoids has been extensively investigated for decades. The reactivities of the vinyl groups of phenylpropanoids are generally low owing to the steric hindrance of their 1,2-disubstituted frameworks. Therefore, the homopolymerization of phenylpropanoids has been a long-standing challenge in this field, and their copolymerization with other reactive monomers, such as styrene and methyl acrylate, has been widely studied[15–18]. Recently, progress was reported in the homopolymerization of phenylpropanoids[19–23], generating polymers with high contents of biomass-derived components. However, even with state-of-the-art polymerization techniques, only a handful of polymers with high phenylpropanoid contents, e.g., >65 wt.% phenylpropanoids, have been reported. In addition, most reported phenylpropanoid-containing polymers exhibit polystyrene-type backbones polymerized via addition at the vinyl moiety, which limits the applicability and recyclability of phenylpropanoids (Fig. 1A). Notably, Labrie-Cleary et al. recently reported the synthesis of photodegradable polymers via

[1]Department of Chemistry and Life Science, Yokohama National University, Yokohama, Japan. [2]Department of Science, University of Toyama, Toyama, Japan. [3]Institute of Advanced Sciences, Yokohama National University, Yokohama, Japan. [4]PRESTO, Japan Science and Technology Agency (JST), Saitama, Japan. [5]These authors contributed equally: Ryo Nagaya, Tatsuya Seko. ✉e-mail: atobe-mahito-wk@ynu.ac.jp; shida-naoki-gz@ynu.ac.jp

the homopolymerization of *o*-hydroxycinnamic acid. However, the resulting material faces limitations in chemical recyclability, as the decomposition products are not readily reusable[23].

To develop decomposable polymers with high phenylpropanoid contents, [2 + 2] cycloaddition reaction represents a promising strategy, as the reactivity of multiply substituted olefins in this transformation is well established[24]. Since [2 + 2] cycloaddition reactions hardly proceed under thermal conditions, they are generally induced via single electron transfer (SET)-triggered reactions using chemical reagents[25–28], electrolysis[29–31], and photoredox catalysis[32–34], and the

photochemical reactions based on energy transfer using photosensitizers (Fig. 1B)[35–37]. Indeed, Bauld et al. reported the [2 + 2] cycloaddition polymerization of diolefins bearing electron-rich aromatic rings via SET-triggered hole-catalytic cycloaddition (Fig. 1B)[38,39]. Although their study introduced a unique polymerization mechanism, the polymers reported by them were neither derived from biomass resources nor designed to be chemically decomposable, thereby limiting their relevance to sustainable polymer development. Regarding the photocatalytic cycloaddition pathway, Oshimura et al. reported the synthesis of a bifunctional cyclobutane-containing monomer from

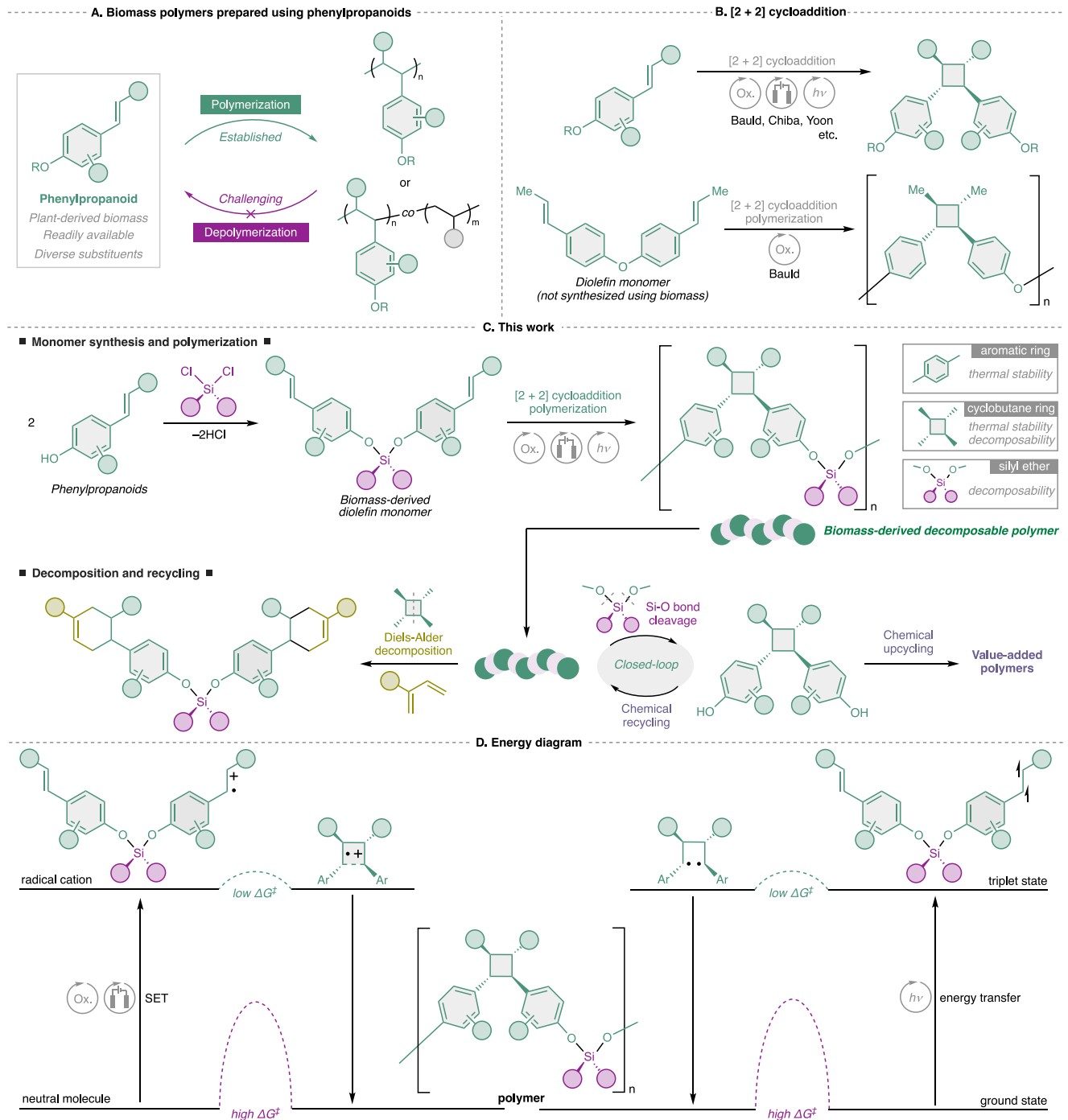

**Fig. 1 | Concept of this work. A** Previous studies regarding the conversion of biomass-derived phenylpropanoids to polymer materials. **B** Precedents of [2 + 2] cycloaddition and polymerization as a method of efficiently polymerizing

phenylpropanoids. **C** Strategy applied in synthesizing biomass-derived decomposable polymers using phenylpropanoids in this study. **D** Energy diagram for [2 + 2] cycloaddition polymerization.

a caffeic acid derivative via [2 + 2] cycloaddition reaction and subsequently reacted it with tetraethylene glycol to produce biomass-derived polyester[40]. Although they have also succeeded in hydrolysis of the resulting polyester, strong acidic or basic conditions and/or heating are typically required for hydrolysis. From the perspective of environmental impact, it seems necessary to design polymers that can be decomposed under milder conditions.

In this study, we report phenylpropanoid-containing polymers with high biomass contents and decomposability via [2 + 2] cycloaddition polymerization (Fig. 1C). Bifunctional monomers containing two phenylpropanoids connected via silyl ether linkages are synthesized, and polymerizations are performed using chemical, electrochemical, and photochemical methods. The polymer design comprises aromatic, cyclobutane, and silyl ether groups in the main chain. The aromatic rings improve the mechanical and thermal properties, whereas the cyclobutane groups provide a high thermal stability due to the thermally unfavorable nature of the retro-[2 + 2] reaction. The Si–O bonds of the silyl ether linkages display higher bond dissociation energies (108 kcal mol$^{-1}$) than that of the $C_{sp3}$–$C_{sp3}$ bond (83 kcal mol$^{-1}$), rendering the polymer chemically and thermally stable. In addition to their robustness, the polymers developed in this study exhibit unique dual decomposition capacities under mild conditions (Fig. 1C). A radical cationic cyclobutane ring is known to be in equilibrium with two olefins, the precursor of the [2 + 2] cycloaddition[32,41]. Therefore, the cyclobutane ring enables a hole-catalytic Diels-Alder reaction in the presence of a diene, enabling an unprecedented mode of polymer decomposition[32]. In addition, the polymers reported herein can be decomposed via reaction with fluoride anions at the Si–O bonds to yield bisphenol products. The resulting bisphenols can be used in chemical re- and upcycling to produce value-added products[42–44]. In general, [2 + 2] cycloaddition reactions between unactivated substrates have high activation energies and do not proceed. However, by activating substrates by one-electron oxidation or energy transfer, the activation energy is reduced, and cycloaddition reaction proceeds (Fig. 1D). Overall, this study proposes a design concept for use in fabricating biomass-derived decomposable polymeric materials with unique main-chain structures, potentially contributing to the sustainable, circular use of carbon resources.

## Results

### Monomer synthesis
First, we attempted to synthesize monomers comprising bisphenylpropanoids connected by silyl ether bonds. The condensation reactions of the phenylpropanoids and dialkyldichlorosilanes were performed at 25 °C in the presence of triethylamine as the base and a catalytic amount of 4-dimethylaminopyridine. We first examined the influences of the substituents on the silicon atom (Table S1). Sterically unhindered substituents lead to destabilization of the resulting monomers, whereas sterically hindered dialkyldichlorosilanes did not react with phenylpropanoids. The reaction between the phenylpropanoids and dichlorodiisopropylsilane proceeded smoothly, affording the desired stable monomers M1–M6 in high yields (Fig. S1).

### Hole-catalytic polymerization using a chemical oxidant
The silyl ether linkages of M1–M6 increase the electron densities in the aromatic rings because of the small electronegativity of the silicon atoms compared to carbon and oxygen atoms, and thus, these monomers may be compatible with the hole-catalytic cycloaddition polymerization initiated via SET. The hole-catalytic cycloaddition polymerization explored by Bauld et al. mostly relied on the use of a catalytic amount of the chemical oxidant magic blue (MB). Thus, we first examined the polymerization of M1 using MB.

MB addition to a solution of M1 resulted in an immediate color change from blue to orange, suggesting a smooth SET between MB and M1. However, reprecipitation afforded a significant amount of an insoluble gel. A small amount of the soluble fraction was subjected to gel permeation chromatography (GPC), which suggested the formation of a polymer of >100 kDa (Fig. S5). Thus, MB induced cross-linking via cationic polymerization at the vinyl groups, in addition to the desired cycloaddition polymerization (Table S3), as observed for specific monomers in a previous study[45].

Inspired by recent progress in hole-catalytic [2 + 2] cycloaddition reactions using hypervalent iodine in small-molecule synthesis[26,46], we changed the oxidant from MB to iodobenzene diacetate (PIDA). M1 was polymerized by adding 50 mol.% PIDA in 1,1,1,3,3,3-hexafluoro-2-propanol (HFIP) at 25 °C (Fig. 2A). After purifying the crude product via reprecipitation, P1 was obtained as a white powder in a 54% yield without the formation of an insoluble product. GPC revealed that the number- ($M_n$) and weight-average molecular weights ($M_w$) of P1 were 3300 and 6200, respectively, and its polydispersity index (PDI) was 1.90 (Fig. 2B). Comparing the $^1$H nuclear magnetic resonance (NMR) spectra of M1 and P1 suggested that the signals representing the olefin protons of M1 were mostly diminished after the reaction. This is concomitant with the manifestation of the signals representing methine protons, which are characteristic of the cyclobutane rings of P1 (Fig. 2C). In addition, the signals representing the terminal olefin protons of P1 were observed at the same positions as those of M1 ($\delta$ = 6.0–6.5). The molecular weight of P1 estimated from the $^1$H NMR was 3600 Da, corresponding to ca. 9.5 repeating units, showing a good agreement with the results of GPC measurement (Fig. S15). From these data, it is suggested that M1 polymerization proceeds via the formation of cyclobutane rings via [2 + 2] cycloaddition to yield P1. The content of biomass-derived ingredients in P1 is 69 wt.%, which is higher than most of the reported polymers synthesized from phenylpropanoid by copolymerization with other non-biomass vinyl monomers.

To gain insight into the polymerization mechanism, the relationship between the conversion of M1 and $M_n$ was investigated. A time-trace experiment revealed that 62% of M1 was consumed within 1 min of PIDA addition (Fig. 2D). Furthermore, $M_n$ exhibited almost no correlation with M1 conversion (Fig. 2E). These data strongly suggest that the reaction proceeded via a chain-growth mechanism with a termination reaction. Intramolecular transfer of a radical cation on the cyclobutane ring after [2 + 2] cycloaddition reaction or back electron transfer from a neutral monomer or polymer would lead to a propagating reaction, but other termination reactions would occur during the reaction. It should be noted that both M1 and P1 can potentially be involved in the propagating reaction by SET, since there was little difference in their oxidation potentials (Fig. S3).

To further investigate the competition between chain propagation and termination reactions, we performed polymerization of M1 with various conditions. The concentration of M1 had minimal impact on the molecular weight of P1 (Tables S4 and S5), whereas increasing the equivalents of PIDA significantly raised the molecular weight (Tables S5–S7). This is presumably because increasing amount of PIDA promoted reoxidation of the polymer chain ends even after swift termination reaction. The plausible mechanism of hole-catalytic polymerization based on these findings is presented in Fig. S10.

Contrasting to the result of M1, the polymerizations of the other monomers (M2–M6) hardly progressed under these conditions (Table S14). Cyclic voltammetry (CV) study of M1–M6 revealed that the oxidation potentials of M2–M6 are higher than that of M1 (Fig. S2), suggesting that the alkene moieties with electron-withdrawing groups, such as ketone or ester, render the monomers less susceptible to SET oxidation. The order of the oxidation potentials of M1–M6 matched with the order of HOMO energy levels estimated by DFT calculations (Table S2).

### Hole-catalytic polymerization via electrolysis
We then investigated electrochemical hole-catalytic cycloaddition polymerization. First, the electrochemical polymerization of M1 was

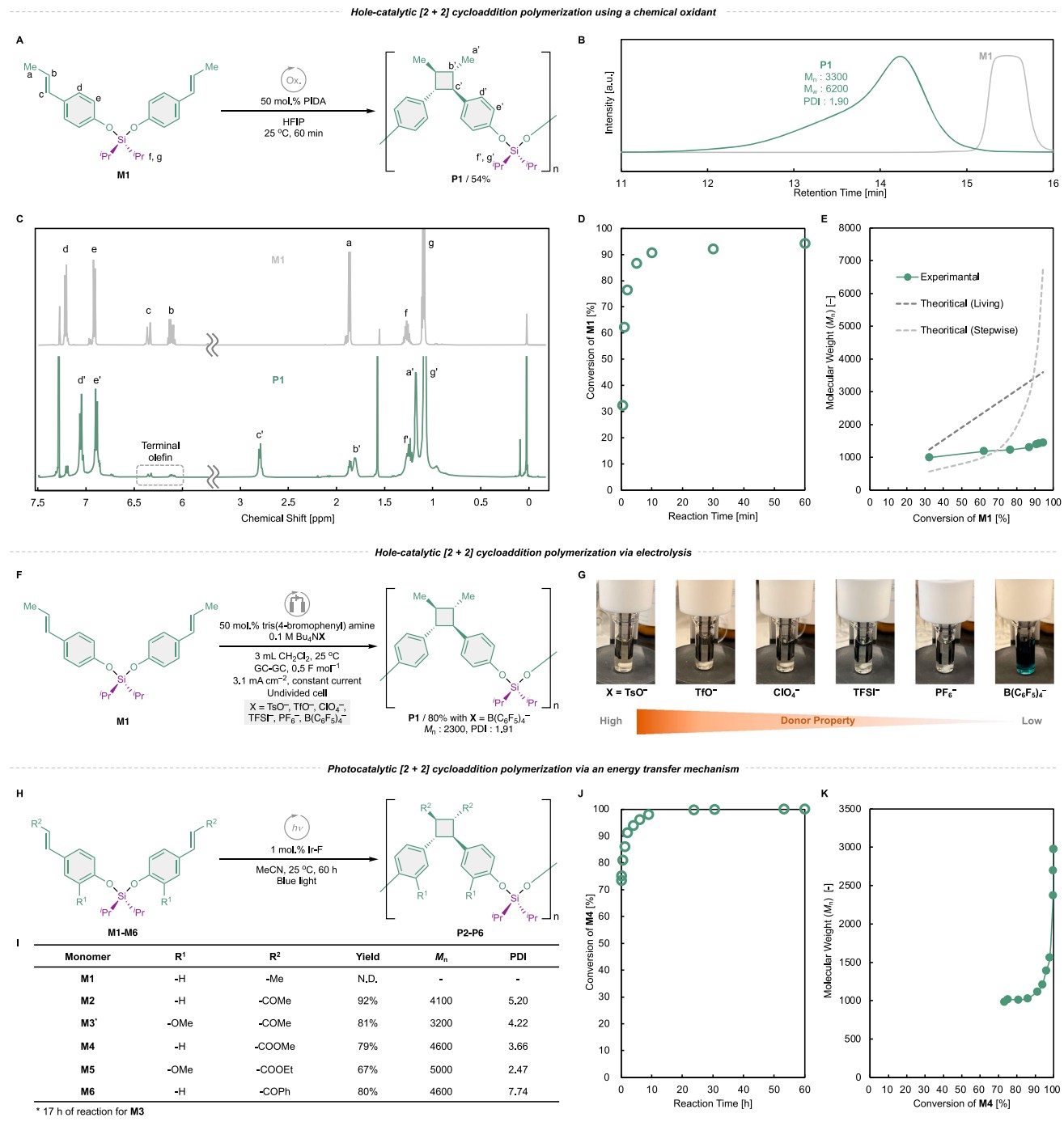

**Fig. 2 | [2 + 2] cycloaddition polymerizations of the monomers. A** Schematic of hole-catalytic polymerization via chemical oxidation using 50 mol.% PIDA as a chemical oxidant. **B** GPC traces (in THF) and **C** $^1$H NMR spectra (in CDCl$_3$) of **M1** and **P1**. **D** Time course of conversion using 0.1 M **M1**, as initiated by PIDA. **E** Relationship between the conversion of **M1** and number-average molecular weight ($M_n$) of **P1**. The green line represents the experimental data, and the dark/light gray dotted lines represent the theoretical lines of living/stepwise polymerization, respectively. **F** Schematic of hole-catalytic polymerization via electrochemical oxidation using

50 mol.% tris(4-bromophenyl) amine as a redox mediator (TsO$^-$ = $p$-toluenesulfonate, TfO$^-$ = trifluoromethanesulfonate, TFSI$^-$ = bis(trifluoromethanesulfonyl) imide). **G** Images of the solutions with different electrolytes during electrolysis. **H** Schematic of photocatalytic [2 + 2] cycloaddition polymerization via energy transfer. **I** Results of photocatalytic [2 + 2] cycloaddition polymerization using the monomers. **J** Time course of conversion using 0.77 M **M4**. **K** Relationship between the conversion of **M4** and $M_n$ of **P4**. Me methyl, Et ethyl, Ph phenyl.

performed via direct oxidation on the anode surface. However, a passivation film was formed immediately, which hampered further electrolysis. To avoid passivation, electrolysis was performed using the redox mediator tris(4-bromophenyl) amine (Fig. 2F). Remarkably, the progress of the polymerization reaction was sensitive to the nature of the anionic species in the supporting electrolyte. The use of the weakly

coordinating anion B(C$_6$F$_5$)$_4$$^-$ was strikingly effective, and the desired polymer was obtained in an 80% yield (Table S13). When Bu$_4$NB(C$_6$F$_5$)$_4$ (Bu = $n$-butyl) was used as the supporting electrolyte, the color of the reaction solution remained blue, which was derived from the radical cationic state of the mediator, whereas more coordinating electrolytes turned the solution orange (Fig. 2G). This suggests that the weakly

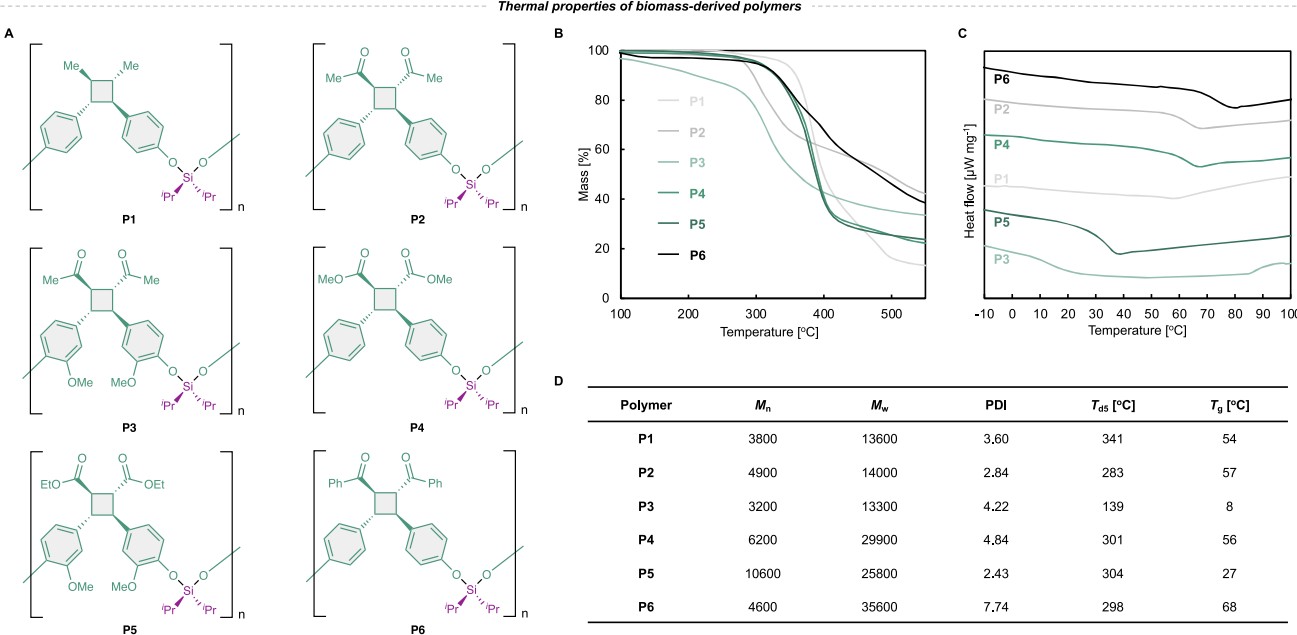

**Fig. 3 | Characteristics of the biomass-derived decomposable polymers.**
**A** Structures of **P1**–**P6**. **B** Thermogravimetric analysis (TGA) thermograms of **P1**–**P6**.
**C** Differential scanning calorimetry (DSC) thermograms of **P1**–**P6**. Heat flow is
shown in the endothermic down direction. **D** Thermal properties of each polymer.

Polymerization conditions, **P1**: 0.096 mM **M1** in HFIP initiated by 100 mol.% PIDA at
25 °C for 60 min. **P2** and **P4**–**P6**: 0.384 M monomer in acetonitrile (MeCN) cata-
lyzed by 1 mol.% of the Ir-F photocatalyst at 25 °C for 60 h. **P3**: 0.384 M **M3** in MeCN
catalyzed by 1 mol.% of the Ir-F photocatalyst at 25 °C for 17 h.

---

coordinating electrolyte solution keeps radical cationic cyclobutane
intact[47], and back electron transfer from the mediator is promoted. It is
noteworthy that the use of tris(4-bromophenyl) amine mediator suc-
cessfully afforded the desired polymer by electrolysis, even though the
aforementioned hole-catalysis by using MB induced uncontrolled
cationic polymerization. This is presumably due to the slow generation
of oxidant by electrochemical reaction was preferable compared to
the rapid addition of oxidant in chemical oxidation reaction.

Meanwhile, the electrochemical polymerizations of **M2**–**M6**
hardly progressed as in the case of chemical oxidants, even in using
tris(2,4-dibromophenyl) amine, which has a higher oxidation potential
than tris(4-bromophenyl) amine (Figs. S2 and S11). This is presumably
due to the lack of nucleophilicity of the monomers and/or difficulty in
the back electron transfer, making it difficult to proceed with a hole-
catalytic [2 + 2] cycloaddition polymerization.

### Photocatalytic polymerization via energy transfer
Several studies report the [2 + 2] cycloaddition of carbonyl-substituted
olefins, including several phenylpropanoids, such as chalcone and
cinnamate, under photocatalysis via an energy transfer
mechanism[35–37]. Inspired by these precedents, we investigated the
polymerizations of **M1**–**M6** under photocatalytic conditions (Fig. 2H).
Blue light was irradiated for 60 h in the presence of 1 mol.% of the (4,4′-
di-*tert*-butyl-2,2′-bipyridine)bis[3,5-difluoro-2-[5-(trifluoromethyl)−2-
pyridinyl]phenyl]iridium(III)     hexafluorophosphate     (Ir-F)    photo-
catalyst. No polymer products were obtained using **M1**, whereas the
polymerizations of **M2**–**M6** proceeded to afford the corresponding
polymers **P2**–**P6** in high yields (Fig. 2I). The biomass contents of the
resulting polymers are 73–79 wt.%. Based on the previous reports, the
successful progress of polymerization for **M2**–**M6** was attributed to
the presence of carbonyl groups adjacent to the olefins, stabilizing the
triplet states[36].

Although the failure of polymerization of **M1** is reasonable due to
the absence of a carbonyl group adjacent to the olefin moiety, the
polymerization of **M1** via hole-catalytic mechanism with Ir-F as a

photoredox catalyst seemed reasonable. However, no polymerization
was observed under these conditions as mentioned above. We attrib-
uted this outcome to the modest oxidizing power of Ir-F ($E_{1/2}$ = +1.21 V
vs. SCE)[48], which is not strong enough to oxidize **M1**. In fact, when we
employed   [Ru(bpz)$_3$][B(C$_6$F$_5$)$_4$]$_2$   (bpz = 2,2′-bipyrazine),   a   photo-
catalyst with a higher oxidation potential ($E_{1/2}$ = +1.45 V vs. SCE)[32,48], **P1**
was successfully obtained via photochemically driven hole-catalytic
polymerization of **M1** (Table S8).

We then investigated the mechanism of photocatalytic poly-
merization using **M4**. The conversion of **M4** exceeded 75% within
10 min of initiating the reaction, indicating a favorable reaction rate
(Fig. 2J, Table S21). In contrast, $M_n$ of **P4** continued to increase even
after the conversion of **M4** had plateaued (Fig. 2K), indicating a
behavior distinct from that observed in the hole-catalytic poly-
merization. This relationship indicates that photocatalytic poly-
merization proceeds via a stepwise mechanism.

**P2**–**P6** showed notably larger PDI values compared to **P1**. This is
presumably due to the difference in polymerization mechanism. Also,
the possibility of forming cyclic polymers by [2 + 2] cycloadditions
between terminal olefins cannot be ruled out. In fact, most of the
terminal olefin protons of photopolymerized polymers are not
observed in the $^1$H NMR spectra.

### Thermal properties
The thermal properties of the resulting polymers were evaluated using
thermogravimetric analysis (TGA) and differential scanning calori-
metry (DSC). Their 5% mass loss temperatures ($T_{d5}$) are in the range
139–341 °C under a nitrogen atmosphere (Fig. 3A, B, D). The $T_{d5}$ gen-
erally depends on the substituents, particularly those on the alicyclic
structures of the main chain, with the following order of thermal sta-
bility: −CH$_3$ (**P1**) > −COO (**P4**, **P5**) > −CO (**P2**, **P3**, **P6**).

The polymers prepared in this study exhibit relatively high glass
transition temperatures ($T_g$) in the range 8–68 °C compared to poly-
mers bearing silyl ether groups (Fig. 3A, C, D). The rigid framework
formed by including the alicyclic structures within the main chain may

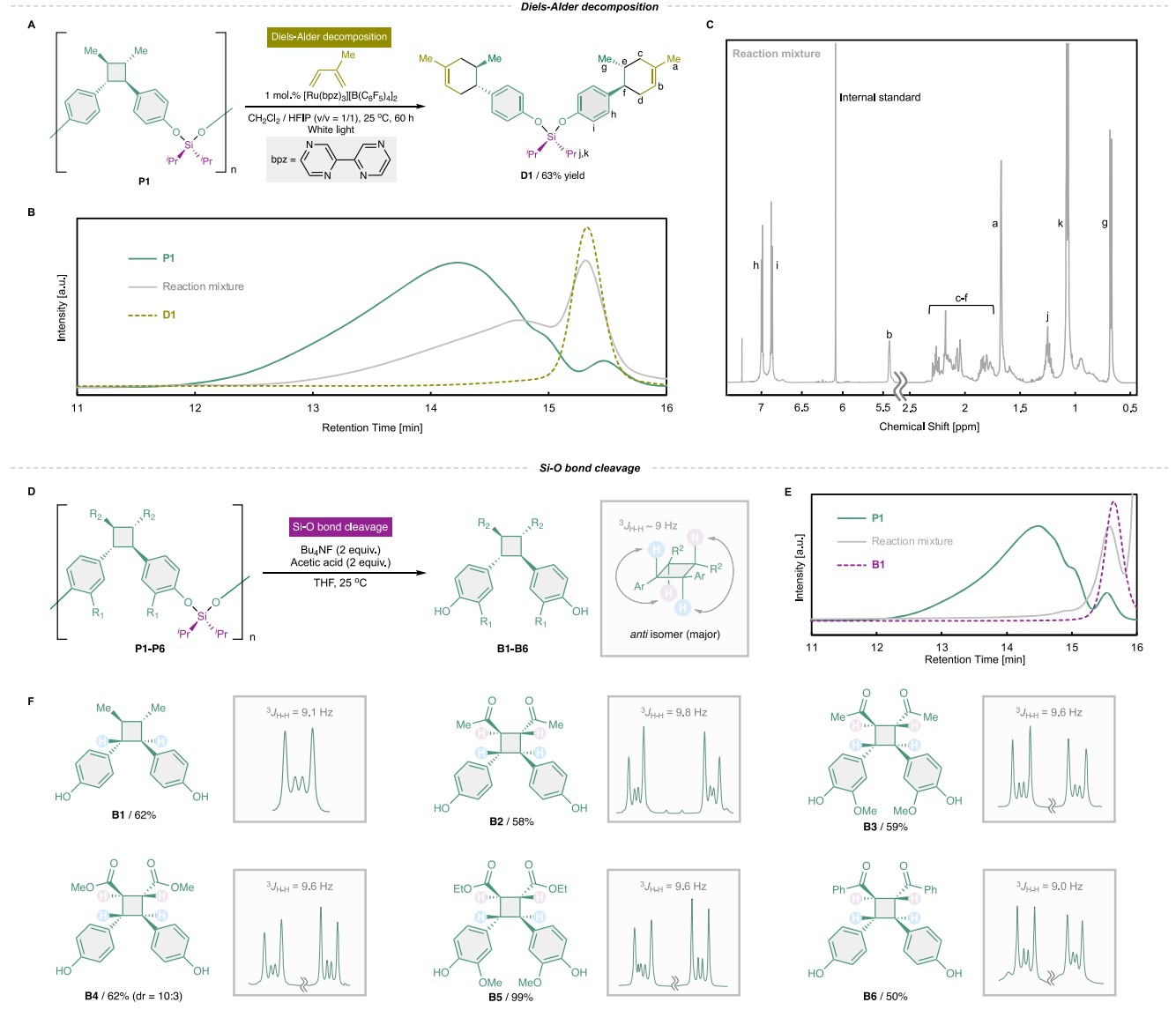

**Fig. 4 | Decomposition reactions of the biomass-derived polymers.** The yields were determined via [1]H NMR spectroscopy of the crude reaction mixtures relative to the internal standard 1,3,5-trimethoxybenzene. Scheme (**A**), GPC traces (in THF) (**B**), and [1]H NMR spectrum (in CDCl$_3$) of the reaction mixture (**C**) of the Diels-Alder decomposition of **P1** using isoprene. Scheme (**D**) and GPC traces (in THF) (**E**) of decomposition via Si−O bond cleavage with Bu$_4$NF, and the [1]H NMR spectra of **B1**−**B6** (**F**). The coupling constants of the adjacent vicinal protons of the methine groups of **B1**−**B6** are noted. THF tetrahydrofuran.

improve $T_g$[49]. In addition, not only the alicyclic structures, but also aromatic rings improve rigidities of the polymers. $T_g$ was the highest for **P6**, which contains the largest number of aromatic rings in its structure. Silyl ether polymers generally display relatively low $T_g$ values owing to their flexible silyl ether bonds with high internal degrees of freedom, and $T_g$ values of <0 °C have been reported[43,44,50]. Additionally, the methoxy group, as a substituent on the aromatic ring, significantly affects $T_g$. The polymers without methoxy groups on their aromatic rings (**P1, P2, P4**, and **P6**) exhibit relatively high $T_g$ values.

The polymers display high thermal stabilities owing to their backbone groups, such as aromatic rings, alicyclic structures, and silyl ether bonds. These results also suggest that the thermal properties of these materials can be tuned by modifying their polymer backbones and substituents. Notably, cyclobutane rings are known for their high distortion energies, and they are thus susceptible to bond cleavage under mechanochemical conditions[51]. The counterintuitively high thermal stabilities should be attributed to the thermally unfavorable

[2 + 2] cycloaddition, and thus, the reverse reaction, i.e., the retro-[2 + 2] reaction, is difficult to proceed thermally either.

## Polymer decomposition

Then, decomposability of **P1**−**P6** was evaluated. First, a decomposition method involving Diels-Alder reactions at the cyclobutane rings was investigated. According to the reaction conditions reported by Yoon et al.[32], [Ru(bpz)$_3$][B(C$_6$F$_5$)$_4$]$_2$ and isoprene were added to reaction solution containing **P1**, and the reaction was performed under white light (Fig. 4A). We have also independently synthesized possible Diels-Alder decomposition product, **D1**, for characterization by GPC and [1]H NMR (see Supplementary Information for details). The GPC trace of the reaction mixture confirmed the decrease in the molecular weight of the polymer, and the formation of small molecular fragment overlapping to **D1** was confirmed (Fig. 4B). [1]H NMR spectroscopy of the crude mixture after the reaction revealed signals representing the target product **D1**. In addition, the peak of **P1** was not observed in

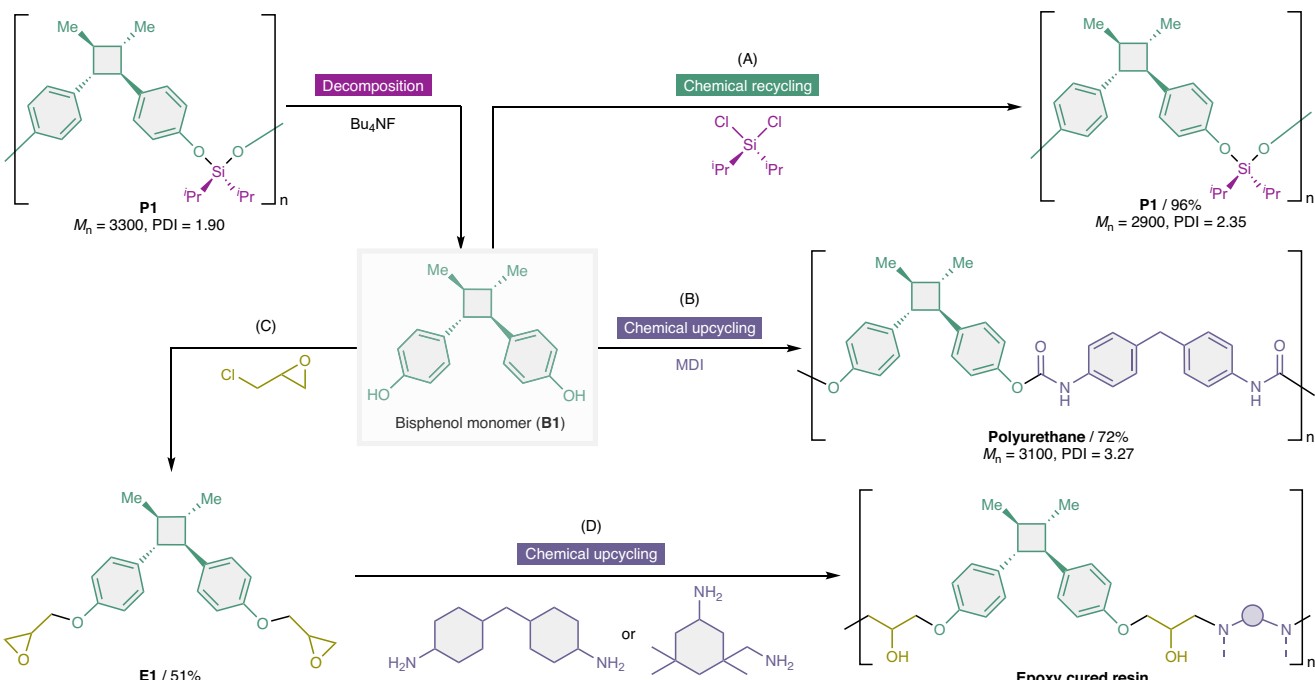

**Fig. 5 | Re- and upcycling of B1. A** Repolymerization to yield **P1** by adding 1 equiv. of dichlorodiisopropylsilane. **B** Synthesis of polyurethane via reaction with 4,4'-methylenebis(phenyl isocyanate) in the presence of 1,4-diazabicyclo[2.2.2]octane. **C** Epoxidation of **B1** via reaction with epichlorohydrin. **D** Curing reactions of epoxide **E1** with curing agents, such as 4,4'-methylenebis(cyclohexylamine) and isophoronediamine, to yield thermoset networks.

the $^1$H NMR spectrum, indicating that the polymer backbone was completely decomposed (Fig. 4C). The calculated $^1$H NMR yield of **D1** was 63% compared to an internal standard. This decomposition approach has an excellent atomeconomic advantage, and the resulting **D1** is expected to be used as an epoxy resin through epoxidation of olefins[52,53].

Subsequently, we performed polymer decomposition involving the cleavage of the Si–O bonds at the linker moieties. Fluoride anions can readily cleave silyl ether bonds under mild conditions, driven by the formation of strong Si–F bonds[42,43,54]. Exploiting this property, the decomposition reactions of **P1–P6** were conducted by adding 2 equiv. of Bu$_4$NF relative to the numbers of silyl ether bonds of the polymers (Fig. 4D). GPC of the reaction solutions detected no high-molecular-weight species, indicating that the polymer backbones were completely decomposed (Figs. 4E and S20). $^1$H NMR spectroscopy confirmed that the decomposition products of **P1–P6** corresponded to the respective bisphenols **B1–B6**, which were produced in high yields (Fig. 4F).

This decomposition also confirmed the regio- and stereo-selectivities of the substituents on the cyclobutane rings. Generally, [2 + 2] cycloadditions produce isomers with different regioselectivities, such as head-to-head or -tail, and stereoselectivities, such as *anti* or *syn*, depending on the reaction method used[55–59]. Regarding the regioselectivity, the coupling pattern of the signals representing the methine protons of the cyclobutane rings changes significantly from head-to-head to head-to-tail. The head-to-head mode exhibits higher-order coupling, whereas the head-to-tail mode does not exhibit such signals[60,61]. The $^1$H NMR spectra of the decomposition products revealed higher-order coupling within **B1–B6**, indicating that head-to-head is the predominant structure of each decomposition product. More significantly, in the case of head-to-head coupling, the coupling constant of the signal representing the methine protons differs depending on the stereochemistry, i.e., *anti* or *syn*. The general coupling constant of the *anti* or *syn* conformation with an adjacent vicinal proton is approximately 9 or 6 Hz, respectively[51,55,56]. The coupling

constants of **B1–B6** were in the range 9.0–9.8 Hz, indicating that the *anti*-isomer is predominant in each case (Fig. 4F). The signals representing head-to-head *syn*-type products were also observed in the $^1$H NMR spectrum of **B4**, and the *anti*:*syn* diastereoselectivity was 10:3. In contrast, detailed stereochemical assignments were challenging across the entire polymer series due to significant signal broadening. Nevertheless, within analyzable regions, we observed vicinal coupling constants of approximately *J* ~ 9 Hz (Fig. S17).

### Chemical re- and upcycling
Biomass-derived bisphenols, which are the products obtained via the cleavage of the linker moieties, display potential for use as bifunctional monomers in chemical re- and upcycling. As a proof-of-concept, **B1** was reacted with a stoichiometric amount of dichlorodiisopropylsilane, and condensation polymerization was performed, resulting in the successful formation of recycled **P1** (Fig. 5A). The molecular weight of recycled **P1** was almost identical to that of the original **P1**, demonstrating that the material can be depolymerized and repolymerized without significant loss of structural integrity. Furthermore, polyurethanes and epoxy-cured products were synthesized using **B1**. The reaction of **B1** with 4,4'-methylenebis(phenyl isocyanate) (MDI) afforded the corresponding polyurethane (Fig. 5B). The reaction of epichlorohydrin with **B1** afforded epoxide **E1** in a 51% yield (Fig. 5C). Subsequently, two types of epoxy-cured resins were synthesized by adding amine curing agents (Fig. 5D). Therefore, the bisphenol products obtained via the decomposition of silyl ether-linked polymers can be easily recycled to form the original polymers and upcycled to generate various value-added polymers.

## Discussion
In this study, phenylpropanoid-derived decomposable polymers with high biomass contents of 69–79 wt.% were successfully developed. By synthesizing bifunctional monomers linked via silyl ethers and employing cycloaddition polymerization, polymers with unique combinations of aromatic, cyclobutane, and silyl ether groups in their

backbones were synthesized. These polymers exhibited excellent thermal properties attributed to their robust structural components. Furthermore, the polymers underwent efficient dual decomposition via Diels-Alder reactions at the cyclobutane rings and Si−O bond cleavage facilitated by fluoride anion, enabling chemical re- and upcycling. This dual decomposability offers a significant advantage in sustainable polymer applications by enabling the decomposition and repurposing of the materials under mild conditions. These findings provide a promising pathway for the development of sustainable polymeric materials using renewable resources, addressing the urgent need to reduce our dependence on petroleum-based polymer materials. This study not only advances the field of biomass-derived polymers but also contributes to environmental sustainability by promoting the circular use of carbon resources. Further increase in molecular weight would be required for practical use, which is currently undergoing in our group. Future research can build on these results to explore further applications and the optimization of these materials.

## Methods

### General procedure for cycloaddition polymerization via chemical oxidation

In a 5 mL vial, a single-electron oxidant dissolved in half the required amount of solvent was added dropwise to the monomer (0.25 mmol) dissolved in the other half of the solvent, while under a nitrogen atmosphere using a balloon. The reaction mixture was stirred for 60 min at a specified temperature and then quenched with methanol. After the solvent was removed under reduced pressure, the crude polymer was purified via reprecipitation (dichloromethane:methanol = 1 mL:20 mL), dissolved in THF at a concentration of 2 mg mL$^{-1}$, and characterized using GPC and NMR.

### General procedure for cycloaddition polymerization via electrochemical oxidation

The reaction was performed in an undivided cell with glassy carbon plates (0.8 × 2 cm) as the anode and cathode, using the ElectraSyn 2.0 Package (IKA, Staufen, Germany). The electrodes were polished using 0.1 and 1 μm alumina and rinsed with deionized water and acetone before the reaction. A solution of the supporting electrolyte was prepared in dichloromethane (0.1 M, 3 mL). After dissolving 0.288 and 0.144 mmol of the monomer and redox mediator, respectively, electrolytic polymerization was performed at room temperature with the application of a constant current. After completion of the reaction, the solvent was removed under reduced pressure. The crude polymer was purified via reprecipitation (dichloromethane:methanol = 1.5 mL: 30 mL), dissolved in THF at a concentration of 2 mg mL$^{-1}$, and characterized using GPC and NMR.

### General procedure for cycloaddition polymerization using the photoredox catalyst

In a 5 mL vial, the photocatalyst dissolved in half the required amount of solvent was added dropwise to the monomer (0.25 mmol) dissolved in the other half of the solvent. The reaction mixture was stirred for a defined period under visible or blue light. After the solvent was removed under reduced pressure, the crude polymer was purified via reprecipitation (**P2**−**P5**, dichloromethane:hexane = 1 mL:20 mL; **P1** and **P6**: dichloromethane:methanol = 1 mL:20 mL), dissolved in THF at a concentration of 2 mg mL$^{-1}$, and characterized using GPC and NMR.

### Procedure for Diels-Alder decomposition

Ru(bpz)$_3$[B(C$_6$F$_5$)$_4$]$_2$ (2.9 mg, 1 mol.% relative to the cyclobutane rings of **P1**), **P1** (57 mg) dissolved in dichloromethane/1,1,1,3,3,3-hexafluoro-2-propanol (HFIP) (v/v = 1/1, 4.7 mL), and isoprene (0.3 mL) were added to a vial bottle, and the reaction mixture was stirred for 60 h under white light. After completion of the reaction, the dichloromethane and

isoprene were removed under reduced pressure. 1,3,5-trimethoxybenzene was then added as an internal standard, and the product was dissolved in deuterated chloroform to calculate the NMR yield of **D1**.

### General procedure for the decomposition at the silyl ether linkage

Two equivalents (relative to the repeating unit of the polymer) of TBAF were added to a solution of the polymer in THF (10 mM repeating units) in the presence of 2 equiv. of acetic acid. The decomposition reaction was conducted by stirring the mixture at room temperature for several hours. After the evaporation of the solvent and vacuum drying, 1 equiv. (relative to the repeating unit of the polymer) of 1,3,5-trimethoxybenzene was added as an internal standard, and the sample was dissolved in the appropriate deuterated solvent. The NMR yields of **B1**−**B6** were determined using the integral value(s) of their methine groups as an index.

## Data availability

The experimental and computational data generated in this study are provided in the article and the Supplementary Information, and are also available from the corresponding authors upon request. Source data are provided with this paper.

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

## Acknowledgements

This work was financially supported by the Japan Society for the Promotion of Science, KAKENHI Grant Nos. 21H05215 (Digi-TOS) to M.A., 23H04916 (Green Catalysis Science) to N.S., 23K17370 to N.S., 23K23386 to N.S., and 24H00394 to M.A., and The Society of Synthetic Organic Chemistry, Japan (DIC Award in Synthetic Organic Chemistry) to N.S., Iketani Science and Technology Foundation to N.S., the Fujimori Science and Technology Foundation to N.S., and Tokuyama Science Foundation to N.S. NMR spectroscopy was performed at the Instrumental Analysis Center (Yokohama National University). The authors thank Ms. Sayaka Kado at the Center for Analytical Instrumentation, Chiba University (Chiba, Japan), for her assistance with high-resolution mass spectrometry. The computation was performed using the Research Center for Computational Science, Okazaki, Japan (Project: 24-IMS-C176 and 25-IMS-C294).

## Author contributions

R.N. and T.S. contributed equally. R.N., M.A., and N.S. wrote the manuscript. R.N., T.S., and N.S. designed and performed all data analyses, R.N., T.S., and K.O. performed synthetic experiments, R.N., T.S., and K.U. performed thermal analysis of the polymers, M.A. and N.S. supervised and directed this project, and all authors contributed to the manuscript.

## Competing interests

The authors declare no competing interests.
