## [Transparent Peer Review file · Nature Communications]

Essential Oil-Derived Decomposable Polymers via Cycloaddition Polymerization of Silyl Ether-Linked Phenylpropanoids

Corresponding Author: Professor Naoki Shida

Version 0:

Reviewer comments:

Reviewer #1

(Remarks to the Author)

The manuscript discusses a novel approach to producing decomposable polymers with high biomass contents via [2 + 2] cycloaddition polymerization using biomass-derived bifunctional monomers, phenyl propanoids with silyl ether linkages. This comprehensive work systematically investigates the polymerization reaction using chemical, electrochemical, and photochemical methods employing a well-defined library of monomers. Although the molecular design and polymerization methods are innovative, the obtained molecular weights of polymers were modest, especially with chemical and electrochemical processes; at the same time, most of the monomers' reactivity was limited. The resulting polymers contained aromatic, cyclobutane rings and silyl ether bonds in their backbones, providing thermal stability and functionality for selective decomposition via Diels-Alder reactions at the cyclobutene rings and fluoride ion addition at Si-O bond cleavage, also providing pathways for the recycling and upcycling. This study comprehensively investigates polymerization, selective depolymerization, and re- and upcycling of high-biomass content decomposable polymers, which could be valuable in achieving a circular economy. The manuscript can be recommended for publication after addressing the following comments:

1. In the case of hole-catalytic polymerization using a chemical oxidant, PIDA, the reaction conversion reached 62% of M1 within 1 minute using 10 mol% of PIDA. This indicates that the reaction rate is swift, enhancing the possibility of a competing termination reaction during chain propagation. Considering such a low molecular weight of P1 (~2300 g/mol), is it possible to investigate the addition of lower equivalents or even sequential aliquots of PIDA?
2. The monomers M2-M6 did not show polymerization in the case of chemical oxidation and electrochemical oxidation. The authors justified this behavior with higher oxidation potential as observed in cyclic voltammograms (CV) in Figure S2; however, a better justification would be needed if 5 of the 6 synthesized monomers do not yield polymerization. The oxidation potential should be identified from CV to calculate the HOMO/LUMO levels, along with DFT calculations, if available.
3. In contrast, M1 could not be polymerized with photocatalysis, which is an ambiguous behavior. This phenomenon requires more explanation.
4. Interestingly, polymer metabolites B1-B6 obtained from Si-O bond breakage showed regioselectivity at methine protons of the cyclobutane rings, favoring head-to-head over head-to-tail. Since these monomers were obtained only from Si-O bond breakage, similar regioselectivity at methine groups could be identified in the polymers, especially when the molecular weights are moderate.
5. Polyurethane, epoxy-cured resin, and even P1 via polycondensation were synthesized during the demonstration of upcycling, which is very promising. The polymer characterization data, such as molecular weights and PDI (GPC traces), should be included.
6. The citation of Figure 1D is missing in the main text; the First citation of Figure S3 is wrongly correlated with GPC traces.

Reviewer #2

(Remarks to the Author)

Reviewer #3

(Remarks to the Author)

This is an interesting article reporting on the synthesis of new, sustainable polymers with good thermal properties, derived from phenylpropanoids — the main component of essential oils — via a [2+2] cycloaddition polymerization process. The monomer design involves introducing a silyl ether linkage between two trans-anole units. The presence of silyl ether linkage in polymers allowed to depolymerize them to monomer or upcycle to valuable products. Meanwhile, the presence of a cyclobutane moiety provides an alternative approach to polymer decomposition, exploiting the Diels–Alder reaction. However, although authors provide us by interesting concept on the synthesis and decomposition of new family of bio-based polymers, the conceptual novelty of this work is questionable. The [2+2] cycloaddition polymerization of quite similar phenylpropanoids (not bio-based) was already reported (see refs. 36, 37). In addition, the values of glass transition temperature of the synthesized polymers are unremarkable, and the decomposition process is usually incomplete. From practical point of view, the usefulness of polymers with such low molecular weight ($M_n < 3,000 \text{ g mol}^{-1}$) as alternative to petroleum-derived plastic is not clear. Therefore, I suggest to reconsider this manuscript after major revision with special emphasis on conceptual novelty of this work.

Some specific comments:

1. Following the decomposition of polymers via silyl ether linkage, the yield of the monomer was reported to be approximately 60%, which is relatively low. An even lower yield was obtained during Diels–Alder decomposition. What is the rest?
2. It is not clear why the molar mass of the synthesized polymers is too low. Why does step-growth polymerization not proceed after the consumption of all monomer, given that polymer has a reactive double bond at the chain ends?
3. Would the addition of new portion of monomer allow to increase the molar mass?
4. Authors should provide by the values of M_n and \bar{D} for polymer (P1) obtained from recycled B1 monomer.
5. In view of the use of column chromatography for the purification of monomers, could the authors comment on the potential scalability of the synthetic protocols developed?
6. Authors claimed that the structure of synthesized polymers provides unique optical and electrical properties. I think it is worth to demonstrate some optical or electrical properties of the synthesized polymers in addition to their thermal properties.

Reviewer #4

(Remarks to the Author)

See attached file.

Version 1:

Reviewer comments:

Reviewer #1

(Remarks to the Author)

The manuscript has been appropriately revised with satisfactory explanations and additional experiments to satisfy the wider audience. According to our suggestion, the authors have carried out polymerization with variation in equivalents of PIDA, calculated the HOMO levels of M2-M6 monomers, achieved photocatalytic polymerization of M1 and assigned stereochemical information of polymers via NMR resonance peaks. I recommend publishing this article without further changes.

Reviewer #2

(Remarks to the Author)

Reviewer #3

(Remarks to the Author)

I fully satisfied with the authors' answers to my questions as well as with the corrections made by authors in the manuscript. Particularly, authors clearly demonstrate the conceptual novelty of their work. In addition, they showed the possibility to synthesize higher molecular weight polymers using their technique. Therefore, I think the manuscript could be accepted for

publication in Nature Commun. in the present form.

Reviewer #4

(Remarks to the Author)

I have reviewed the content and can confirm that all previously raised concerns and suggested improvements have been appropriately addressed.

I have no further comments and find the content acceptable as is.

Reviewer #1

General comment:

The manuscript discusses a novel approach to producing decomposable polymers with high biomass contents via [2 + 2] cycloaddition polymerization using biomass-derived bifunctional monomers, phenyl propanoids with silyl ether linkages. This comprehensive work systematically investigates the polymerization reaction using chemical, electrochemical, and photochemical methods employing a well-defined library of monomers. Although the molecular design and polymerization methods are innovative, the obtained molecular weights of polymers were modest, especially with chemical and electrochemical processes; at the same time, most of the monomers' reactivity was limited. The resulting polymers contained aromatic, cyclobutane rings and silyl ether bonds in their backbones, providing thermal stability and functionality for selective decomposition via Diels-Alder reactions at the cyclobutene rings and fluoride ion addition at Si–O bond cleavage, also providing pathways for the recycling and upcycling. This study comprehensively investigates polymerization, selective depolymerization, and re- and upcycling of high-biomass content decomposable polymers, which could be valuable in achieving a circular economy. The manuscript can be recommended for publication after addressing the following comments:

Response: We sincerely thank the reviewer for their thorough and thoughtful evaluation of our manuscript. We greatly appreciate the recognition of the novelty and comprehensiveness of our approach, including the design of biomass-derived bifunctional monomers and the systematic exploration of chemical, electrochemical, and photochemical polymerization pathways. We also fully acknowledge the reviewer's concerns regarding the relatively modest molecular weights and the limited reactivity observed for several monomers. In response, we have conducted additional experiments, refined our mechanistic discussion, and revised relevant sections in both the main text and Supporting Information to clarify these issues. We believe these revisions have significantly strengthened the manuscript and improved its clarity. We are grateful for the reviewer's constructive feedback, which has been invaluable in guiding these improvements and enhancing the overall quality of the work.

Comment: 1. In the case of hole-catalytic polymerization using a chemical oxidant, PIDA, the reaction conversion reached 62% of M1 within 1 minute using 10 mol% of PIDA. This indicates that the reaction rate is swift, enhancing the possibility of a competing termination reaction during chain propagation. Considering such a low molecular weight of P1 (~2300 g/mol), is it possible to investigate the addition of lower equivalents or even sequential aliquots of PIDA?

Response: We appreciate this insightful suggestion. As advised, we conducted additional polymerization experiments using lower equivalents of PIDA (1.25–5.0 mol%) as well as sequential aliquot additions (3.3 mol% × 3, added at 10-minute intervals). However, these approaches did not result in an increase in the molecular weight of **P1** (Table S6, Entries 1–4). This observation suggests that reducing the amount of PIDA limits not only the termination step but also the chain propagation, leading to insufficient chain growth overall. In contrast, increasing the PIDA loading led to a significant increase in the molecular weight of **P1** (Table S6, Entries 6–8). A similar trend was observed when the reaction was conducted in HFIP as the sole solvent (Table S7), and the molecular weight of **P1** increased compared to that obtained using HFIP/CH₂Cl₂ co-solvent system. This enhancement is likely due to the improved

stabilization of the radical cation intermediate in HFIP. These data are now included in the revised manuscript (Figure 2A and 2B) and detailed in the revised Supporting Information (Tables S5–S8).

Table S6. The effect of PIDA equivalent on hole-catalytic polymerization using HFIP/CH₂Cl₂ co-solvent

Entry	Amount of PIDA [mol.%]	Yield of P1 [%]	M_n	M_w	PDI
1	1.25	41	1200	1900	1.58
2	2.5	46	1500	2800	1.84
3	5	60	1800	3900	2.22
4	3.3×3^a	81	1900	4600	2.47
5	10	79	2400	4600	1.94
6	30	77	2200	5100	2.30
7	50	61	2200	5200	2.40
8	100	14	2000	3800	1.91

a) PIDA was added in three portions at 10 min intervals.

Table S7. The effect of PIDA equivalent on hole-catalytic polymerization using HFIP

Entry	Amount of PIDA [mol.%]	Yield of P1 [%]	M_n	M_w	PDI
1	10	75	2400	3700	1.57
2	30	76	2800	5000	1.76
3	50	54	3300	6200	1.90
4	100	25	3800	13600	3.60

Figure 2. [2 + 2] cycloaddition polymerizations of the monomers. (A) Schematic of hole-catalytic polymerization via chemical oxidation using 50 mol.% PIDA as a chemical oxidant. (B) GPC traces and (C) ¹H NMR spectra of **M1** and **P1**. (D) Time course of conversion using 0.1 M **M1**, as initiated by PIDA. (E) Relationship between the conversion of **M1** and number-average molecular weight (M_n) of **P1**. The green line represents the experimental data, and the dark/light gray dotted lines represent the theoretical lines of living/stepwise polymerization, respectively. (F) Schematic of hole-catalytic polymerization via electrochemical oxidation using 50 mol.% tris(4-bromophenyl) amine as a redox mediator (TsO⁻ = *p*-toluenesulfonate, TfO⁻ = trifluoromethanesulfonate, TFSI⁻ = bis(trifluoromethanesulfonyl)imide). (G) Images of the solutions with different electrolytes during electrolysis. (H) Schematic of photocatalytic [2 + 2] cycloaddition polymerization via energy transfer. (I) Results of photocatalytic [2 + 2] cycloaddition polymerization using the monomers. (J) Time course of conversion using 0.77 M **M4**. (E) Relationship between the conversion of **M4** and M_n of **P4**. Me, methyl; Et, ethyl; Ph, phenyl.

Comment: 2. The monomers M2–M6 did not show polymerization in the case of chemical oxidation and electrochemical oxidation. The authors justified this behavior with higher oxidation potential as observed in cyclic voltammograms (CV) in Figure S2; however, a better justification would be needed if 5 of the 6 synthesized monomers do not yield polymerization. The oxidation potential should be identified from CV to calculate the HOMO/LUMO levels, along with DFT calculations, if available.

Response: We agree with the reviewer that a deeper rationale is necessary to explain why only M1 successfully underwent polymerization while M2–M6 did not. In response, we performed DFT calculations to determine the HOMO energy levels of M1–M6, which help assess their susceptibility to oxidation. The calculated HOMO levels correlated well with the oxidation potentials obtained from cyclic voltammetry (Figure S2), confirming that M1 possesses the highest HOMO level among the series and is thus the most easily oxidized. This result supports our interpretation that the hole-catalytic polymerization is governed by the monomer’s oxidation potential. We have added the HOMO data to Table S2 and expanded the discussion in the revised manuscript to strengthen the mechanistic rationale.

Table S2. HOMO energies of M1–M6 identified from DFT calculation

Entry	Monomer	Energy of HOMO [eV]
1	M1	-7.15
2	M2	-7.74
3	M3	-7.41
4	M4	-7.82
5	M5	-7.31
6	M6	-7.70

Figure S2. Cyclic voltammograms of M1–M6, tris(4-bromophenyl) amine, and tris(2,4-dibromophenyl) amine.

Comment: 3. In contrast, **M1** could not be polymerized with photocatalysis, which is an ambiguous behavior. This phenomenon requires more explanation.

Response: We thank the reviewer for pointing out this important issue. Photocatalytic [2 + 2] cycloaddition reactions of phenylpropanoid derivatives, such as chalcones, are typically enabled by energy transfer mechanisms involving excited triplet states. This reactivity is often attributed to the presence of carbonyl groups, which stabilize the triplet states. In contrast, **M1** lacks a carbonyl moiety, and thus its failure to undergo photocatalytic polymerization is likely due to its poor ability to access a reactive triplet state. Alternatively, we considered the possibility that the Ir-based photocatalyst Ir-F might act as a one-electron oxidant to initiate hole-catalytic polymerization of **M1**. However, no polymerization was observed under these conditions. This is due to the weak oxidizing power of Ir-F ($E_{1/2} = +1.21$ V vs. SCE; Akita *et al.*, *Inorg. Chem. Front.* **2014**, *1*, 564.), which is not strong enough to oxidize **M1**. To test this hypothesis, we employed $[\text{Ru}(\text{bpz})_3][\text{B}(\text{C}_6\text{F}_5)_4]_2$, a photocatalyst with a higher oxidation potential ($E_{1/2} = +1.45$ V vs. SCE; Yoon *et al.*, *Chem. Sci.* **2012**, *3*, 2807), and successfully achieved photochemically driven hole-catalytic polymerization of **M1** (Table S8). These findings, along with the cited references and new experimental data, have been incorporated into the revised manuscript and Supporting Information.

Table S8. Hole-catalytic polymerization of **M1** using a photocatalyst

Entry	Photocatalyst	Solvent	Light	Yield [%]	M_n	PDI
1	Ir-F	MeCN	Blue	N.D.	-	-
2	$[\text{Ru}(\text{bpz})_3][\text{B}(\text{C}_6\text{F}_5)_4]_2$	$\text{CH}_2\text{Cl}_2/\text{HFIP}$ (v/v = 1/1)	White	40	2500	3.51
3	$[\text{Ru}(\text{bpz})_3][\text{B}(\text{C}_6\text{F}_5)_4]_2$	HFIP	White	56	3600	4.06

Oxidizing powers of Ir-F and $[\text{Ru}(\text{bpz})_3][\text{B}(\text{C}_6\text{F}_5)_4]_2$ are 1.21 and 1.45 V vs. SCE, respectively.

Comment: 4. Interestingly, polymer metabolites B1-B6 obtained from Si-O bond breakage showed regioselectivity at methine protons of the cyclobutane rings, favoring head-to-head over head-to-tail. Since these monomers were obtained only from Si-O bond breakage, similar regioselectivity at methine groups could be identified in the polymers, especially when the molecular weights are moderate.

Response: We agree that this is an important and insightful observation. As suggested, we examined the ^1H NMR spectra of the polymers to evaluate the methine proton environments in the cyclobutane rings. However, due to significant signal broadening, detailed stereochemical

assignments were challenging across the entire polymer series. Nevertheless, within analyzable regions, we observed vicinal coupling constants of approximately $J \sim 9$ Hz (Figure S17). This coupling constant is consistent with that observed for the head-to-head isomeric structures in the decomposition products (Figure 4F), supporting the hypothesis that similar regioselectivity is preserved in the polymers. These findings have now been incorporated into the revised manuscript and Supporting Information to strengthen the discussion of stereochemical features.

Figure S17. The coupling constants of the adjacent vicinal protons of methine groups of **P1** and **P2**

Comment: 5. Polyurethane, epoxy-cured resin, and even P1 via polycondensation were synthesized during the demonstration of upcycling, which is very promising. The polymer characterization data, such as molecular weights and PDI (GPC traces), should be included.

- **Response:** Thank you for this valuable suggestion. In response, we have clearly presented the molecular weight data (M_n and PDI) for the recycled **P1** and the polyurethane product. These values are now included in the main manuscript (Figure 5), and the corresponding GPC traces are provided in the Supporting Information under section 3-5-3 “*Recycling and Upcycling of BI*”.

In addition, the general synthetic procedures and molecular characterization details for each upcycled product are described in section 2-4 of the Supporting Information.

Please note that molecular weight data and GPC traces could not be obtained for the epoxy-cured resin due to its crosslinked, insoluble nature, which prevents dissolution in any solvent suitable for GPC analysis.

These updates ensure that all available characterization data for the upcycled materials are now explicitly reported.

Figure 5. Re- and upcycling of **B1**. (A) Repolymerization to yield **P1** by adding 1 equiv. of dichlorodiisopropylsilane. (B) Synthesis of polyurethane via reaction with 4,4'-methylenebis(phenyl isocyanate) in the presence of 1,4-diazabicyclo[2.2.2]octane. (C) Epoxidation of **B1** via reaction with epichlorohydrin. (D) Curing reactions of epoxide **E1** with curing agents, such as 4,4'-methylenebis(cyclohexylamine) and isophoronediamine, to yield thermoset networks.

Comment: 6. The citation of Figure 1D is missing in the main text; the first citation of Figure S3 is wrongly correlated with GPC traces.

- **Response:** We apologize for this oversight and appreciate the reviewer's careful attention to detail. We have now properly cited Figure 1D at the appropriate place in the revised manuscript. In addition, the erroneous reference to Figure S3 has been corrected: the GPC traces are now accurately cited as Figure S5. These citation errors have been fully addressed in the revised version.

Reviewer #2

General comment: I co-reviewed this manuscript with one of the reviewers who provided the listed reports. This is part of the Nature Communications initiative to facilitate training in peer review and to provide appropriate recognition for Early Career Researchers who co-review manuscripts.

- **Response:** We thank Reviewer 2 for their co-review of our manuscript as part of the peer review training initiative. We have addressed all points raised in the co-reviewed report and welcome any additional suggestions or clarifications.

Reviewer #3

General comment:

This is an interesting article reporting on the synthesis of new, sustainable polymers with good thermal properties, derived from phenylpropanoids — the main component of essential oils — via a [2+2] cycloaddition polymerization process. The monomer design involves introducing a silyl ether linkage between two trans-anole units. The presence of silyl ether linkage in polymers allowed to depolymerize them to monomer or upcycle to valuable products. Meanwhile, the presence of a cyclobutane moiety provides an alternative approach to polymer decomposition, exploiting the Diels–Alder reaction. However, although authors provide us by interesting concept on the synthesis and decomposition of new family of bio-based polymers, the conceptual novelty of this work is questionable. The [2+2] cycloaddition polymerization of quite similar phenylpropanoids (not bio-based) was already reported (see refs. 36, 37). In addition, the values of glass transition temperature of the synthesized polymers are unremarkable, and the decomposition process is usually incomplete. From practical point of view, the usefulness of polymers with such low molecular weight ($M_n < 3,000 \text{ g mol}^{-1}$) as alternative to petroleum-derived plastic is not clear. Therefore, I suggest to reconsider this manuscript after major revision with special emphasis on conceptual novelty of this work.

- **Response:** We thank the reviewer for their thoughtful and critical evaluation of our manuscript. We appreciate the recognition of the chemical design, including the use of silyl ether and cyclobutane motifs for decomposition and upcycling. We understand the reviewer's concern regarding the conceptual novelty, especially in light of prior studies involving [2 + 2] cycloaddition polymerization of phenylpropanoid derivatives. To address this, we have revised the *Introduction* and *Results and Discussion* sections to clearly articulate the differences between our biomass-derived monomer system and the previously reported non-renewable systems, emphasizing the integration of high biomass content, dual decomposition modes, and upcycling strategies that have not been concurrently demonstrated in earlier works. Regarding the relatively low molecular weights and modest T_g values of the polymers, we fully acknowledge these limitations. In the revised manuscript, we have added a more cautious interpretation of these results, while also emphasizing the synthetic accessibility, chemical versatility, and decomposition/upcycling capabilities of our materials as key attributes toward sustainable materials development. We believe that these clarifications and additions improve the rigor and clarity of our manuscript, and we hope the reviewer will find that our revisions adequately address the concerns raised.

Comment: 1. Although the authors provide an interesting concept, the conceptual novelty is questionable given that [2+2] cycloaddition polymerization of similar (non-biomass) phenylpropanoids has been reported (refs. 36, 37).

- **Response:** We respectfully acknowledge the reviewer's concern regarding conceptual novelty. Indeed, Bauld *et al.* reported [2 + 2] cycloaddition polymerization of non-renewable aromatic diolefin monomers (N. L. Bauld *et al.*, *Macromolecules*, **1996**, *29*, 3661-3662., and N. L. Bauld *et al.*, *J. Phys. Org. Chem.*, **1999**, *12*, 808-818.). However, our work represents the first demonstration of [2 + 2] cycloaddition polymerization using biomass-derived bifunctional phenylpropanoids, which achieve a high biomass content of up to 79 wt%. Furthermore, unlike

the polymers reported by Bauld *et al.*, which were non-decomposable, our polymers incorporate silyl ether and cyclobutane functionalities that enable selective chemical decomposition, recycling, and upcycling.

In addition, the stereochemical configuration of the cyclobutane moieties has been elucidated in our study, whereas Bauld's work did not report such structural details. Taken together, these distinctions in monomer origin, polymer functionality, decomposability, and stereochemical insight collectively underscore the conceptual novelty and broader significance of our approach.

Comment: 2. The decomposition yield of monomers via silyl ether cleavage is ~60%, and even lower for Diels-Alder. What is the rest?

- **Response:** We thank the reviewer for this important question. The observed decomposition yields indicate that not all monomeric units could be recovered in a defined form. A plausible explanation is that the liberated monomers or intermediate species undergo secondary decomposition, possibly due to oxidative decomposition under ambient conditions. To investigate this, we monitored the crude reaction mixture by GC-MS over time. Several new byproducts appeared in the chromatogram a few days after the decomposition reaction (Figure S21), which were absent in the immediately analyzed sample. Unfortunately, these byproducts could not be structurally identified based on their m/z values, likely due to fragmentation or overlapping signals. Nevertheless, this observation supports the hypothesis that the remaining mass balance is attributed to further transformation of the initial decomposition products.

Figure S21. GC-MS chromatogram of the crude mixtures immediately after (top) and several days after (bottom) the decomposition reaction of **P1**.

Comment: 3. It is not clear why the molar mass of the synthesized polymers is too low. Why does step-growth polymerization not proceed after monomer consumption, given reactive terminal alkenes?

- Response:** This is an important mechanistic question, and we thank the reviewer for raising it. Regarding the hole-catalytic polymerization using PIDA, we observed that the molecular weight of **P1** plateaued within approximately 10–15 minutes after initiation (Figures 2D, 2E). This suggests that premature chain termination may be competing with propagation, thereby limiting the achievable molecular weight. To test this hypothesis, we increased the amount of PIDA to promote reoxidation of the terminated species and indeed observed a measurable increase in molecular weight (Tables S6 and S7). We also performed polymerization using isolated low-molecular-weight **P1** as the monomer and found a modest increase in molecular weight (Scheme S1), which further supports the possibility of limited propagating reaction due to inefficient reinitiation. On the other hand, photocatalytic polymerization via an energy transfer mechanism proceeded via a stepwise process. In this case, **P3** showed especially small molecular weight, presumably due to the short reaction time (Figure S15, entry 6). Instead, we

prolonged the reaction time for photocatalytic polymerization of **M3**, and successfully obtained **P3** with a higher molecular weight (Figure S15, entry 7).

Table S6. The effect of PIDA equivalent on hole-catalytic polymerization using HFIP/CH₂Cl₂ co-solvent

Entry	Amount of PIDA [mol.%]	Yield of P1 [%]	M_n	M_w	PDI
1	1.25	41	1200	1900	1.58
2	2.5	46	1500	2800	1.84
3	5	60	1800	3900	2.22
4	3.3×3^a	81	1900	4600	2.47
5	10	79	2400	4600	1.94
6	30	77	2200	5100	2.30
7	50	61	2200	5200	2.40
8	100	14	2000	3800	1.91

a) PIDA was added in three portions at 10 min. intervals.

Table S7. The effect of PIDA equivalent on hole-catalytic polymerization using HFIP

Entry	Amount of PIDA [mol.%]	Yield of P1 [%]	M_n	M_w	PDI
1	10	75	2400	3700	1.57
2	30	76	2800	5000	1.76
3	50	54	3300	6200	1.90
4	100	25	3800	13600	3.60

Scheme S1. Hole-catalytic polymerization using **P1** as a raw material

Table S15. Polymerization of **M3** using various photocatalysts

Entry	Concentration [M]	Photocatalyst	Time [h]	Yield [%]	M_n	M_w	PDI
1	0.384	Acridinium Perchlorate ^{a)}	12	7	1100	1100	1.08
2 ^{b)}	0.384	[Ru(bpz) ₃][PF ₆] ₂	12	3	1100	1300	1.19
3	0.384	9-Fluorenone ^{c)}	30	48	1400	1800	1.31
4	0.192	Ir-F	12	59	1700	2400	1.40
5	0.384	Ir-F	1	trace	-	-	-
6	0.384	Ir-F	12	79	3000	6900	2.34
7	0.384	Ir-F	17	81	3200	13300	4.22
8	0.384	Ir-F	30	Unmeasurable ^{d)}	-	-	-
9	0.384	Ir-F	60	Unmeasurable ^{d)}	-	-	-
10	0.768	Ir-F	12	74	2100	3500	1.66

Ir-F = (4,4'-Di-tert-butyl-2,2'-bipyridine)bis[3,5-difluoro-2-[5-(trifluoromethyl)-2-pyridinyl]phenyl]iridium(III) Hexafluorophosphate.

a) 9-mesityl-10-methylacridinium perchlorate. b) Visible light. c) Photocatalyst (10 mol.%). d) Gelatinous components, which are insoluble in any solvent, are formed. MeCN, acetonitrile.

Comment: 4. Would the addition of a new portion of monomer allow to increase the molar mass?

- Response:** We are grateful to the reviewer for this insightful suggestion, which provided a new perspective on the polymerization mechanism. To evaluate this possibility, we conducted a series of experiments in which additional portions of **M1** were introduced during the polymerization reaction. However, as shown in Table S5, Entry 2, the molecular weight of **P1** did not increase upon monomer addition alone. In contrast, when additional PIDA was added during the reaction (Table S5, Entry 3), the molecular weight increased, suggesting that chain growth was primarily limited by the availability of oxidant. Moreover, when we added both **M1** and PIDA together (Table S5, Entry 4), the molecular weight did not further improve, reinforcing the conclusion that the molecular weight of **P1** is more strongly influenced by the equivalent amount of PIDA than by the monomer concentration. These findings support the

notion that efficient reoxidation of propagating species is critical for achieving higher degrees of polymerization in this system. We have clarified this mechanistic insight in the revised manuscript.

Table S5. The effect of adding extra PIDA and/or **M1** on molecular weight of **P1**

Entry	Initial condition		Variation from the initial condition		Yield of P1 [%]	M_n	M_w	PDI
	M1 [mmol]	PIDA [mol.%]	M1 [mmol]	PIDA [mol.%]				
1	0.25	10	0.25	10	79	2400	4600	1.94
2 ^a	0.19	10	0.19 → 0.40	10 → 5	— ^c	1400	3500	2.57
3 ^b	0.25	3.3	0.25	3.3 → 10	81	1900	4600	2.47
4 ^a	0.19	3.3	0.19 → 0.55	3.3	— ^c	1700	3900	2.36

M1 and/or PIDA were added in three portions at a) 1 min., b) 10 min. intervals. c) It was not possible to determine the exact yields due to the large amount of monomer remaining.

Comment: 5. Could the authors comment on the scalability of monomer synthesis, given the use of column chromatography?

- Response:** We agree with the reviewer that the scalability of monomer synthesis is a critical consideration for future practical applications. The use of column chromatography, while convenient at the laboratory scale, would indeed increase economic and operational costs for large-scale production. However, we note that the synthesis of the monomers involves highly selective condensation reactions between dichlorosilanes and phenols, which predominantly generate easily separable side products. For liquid monomers, purification can generally be accomplished by simple extraction techniques, while solid monomers can be purified by recrystallization, without the need for chromatography. We have confirmed that monomers purified via these simpler procedures are sufficiently pure for polymerization to proceed efficiently. Therefore, we believe that the monomer synthesis can be adapted for larger-scale preparation without relying on column chromatography.

Comment: 6. The authors claimed unique optical/electrical properties, but no data are shown.

- Response:** We appreciate the reviewer's observation. We agree that the mention of optical and electrical properties was speculative, as we did not present any supporting data. Therefore, to maintain clarity and focus, we have removed such claims from the main text and now limit our discussion to experimentally demonstrated properties, namely the thermal stability and

chemical recyclability of the synthesized polymers. We believe this revision aligns the manuscript more closely with its central contribution—introducing a conceptually new class of chemically recyclable polymers derived from biomass.

Reviewer #4

General comment:

In this article, the authors showed a sustainable method for the synthesis of high-biomass decomposable polymers via [2+2] cycloaddition polymerization. The polymers obtained exhibited excellent thermal properties and underwent efficient dual decomposition via Diels-Alder reactions and Si-O bond cleavage. The schemes and figures were easy to understand, there were no careless mistakes, and the paper felt very easy to read. The concept of this research topic is intriguing. However, there are still several hurdles to overcome before it can be published. The revision is needed before acceptance for publishing in "Nature Communications".

- **Response:** We sincerely thank the reviewer for their positive assessment of our manuscript. We have thoroughly revised the manuscript to address the reviewer's comments, including the addition of new data, clarification of mechanistic insights, and a more critical discussion of limitations. We believe these revisions have substantially improved the manuscript and we are grateful for the opportunity to refine our work.

Comment: 1. At the end of the second paragraph of their three-page introduction, the authors write: Moreover, to the best of our knowledge, no studies regarding decomposable polymers prepared using phenylpropanoids, which are desirable for the sustainability of the polymer industry, have been reported (Figure 1A).

However, a paper with a similar concept was published in 2024 with the following title:

Synthesis of photodegradable polyesters from bio-based 3,4-dimethoxycinnamic acid and investigation of their degradation behaviors (DOI: 10.1016/j.polymer.2024.127204)

As this paper was found through a quick search, the authors should conduct a more careful literature search and cite papers for comparison where necessary.

- **Response:** We thank the reviewer for bringing this relevant publication by Oshimura *et al.* to our attention. We have now carefully reviewed the cited study and agree that it shares the broader goal of developing decomposable polymers from phenylpropanoid-derived monomers. That said, there are several key differences between the two approaches. The Oshimura's study focuses on polyesters derived from 3,4-dimethoxycinnamic acid and relies on conventional ester hydrolysis for depolymerization, which typically requires strong acidic or basic conditions and/or heating. In contrast, our work expands the structural diversity of phenylpropanoid-based monomers by introducing silyl ether linkages, enabling decomposition through orthogonal and mild stimuli (fluoride ion or Diels-Alder reaction) even in thermally stable systems. These features allow us to demonstrate a dual and selective decomposition strategy under relatively mild conditions, which we believe represents a distinct and academically novel contribution to the field of sustainable polymer chemistry.

We have revised the introduction accordingly to include this reference and clarify the conceptual and mechanistic distinctions between our study and the report by Oshimura *et al.*

Comment: 2. The molecular weights of the polymers synthesized this time are generally low, with some having Mw below 4000. It seems difficult to compare the thermal properties of TGA and DSC at this molecular weight and say that it is the effect of a particular functional group.

- **Response:** We thank the reviewer for raising this important point. As noted, some of the polymers initially used for thermal property evaluation (particularly **P1** and **P3**) had relatively low molecular weights ($M_w < 4000$), which may have complicated the interpretation of structure–property relationships.

To address this concern, we conducted additional polymerizations to obtain higher molecular weight samples of both **P1** and **P3**. High- M_w **P1** was synthesized by polymerizing **M1** with 1.0 equivalent of PIDA in HFIP (Table S7, Entry 4), and high- M_w **P3** was synthesized by photocatalytic polymerization of **M3** for 17 hours (Table S15, Entry 7).

We then re-evaluated the thermal properties (T_{d5} and T_g) of these high- M_w samples and found that the trend remained consistent with the original data, supporting the conclusion that the observed differences in thermal behavior arise from structural variations among the polymers, rather than from differences in molecular weight.

We have included these new data and discussion in the revised manuscript and Supporting Information (Figure 3, Table S15).

Table S7. The effect of PIDA equivalent on hole-catalytic polymerization using HFIP

Entry	Amount of PIDA [mol.%]	Yield of P1 [%]	M_n	M_w	PDI
1	10	75	2400	3700	1.57
2	30	76	2800	5000	1.76
3	50	54	3300	6200	1.90
4	100	25	3800	13600	3.60

Table S15. Polymerization of **M3** using various photocatalysts

Entry	Concentration [M]	Photocatalyst	Time [h]	Yield [%]	M_n	M_w	PDI
1	0.384	Acridinium Perchlorate ^{a)}	12	7	1100	1100	1.08
2 ^{b)}	0.384	[Ru(bpz) ₃][PF ₆] ₂	12	3	1100	1300	1.19
3	0.384	9-Fluorenone ^{c)}	30	48	1400	1800	1.31
4	0.192	Ir-F	12	59	1700	2400	1.40
5	0.384	Ir-F	1	trace	-	-	-
6	0.384	Ir-F	12	79	3000	6900	2.34
7	0.384	Ir-F	17	81	3200	13300	4.22
8	0.384	Ir-F	30	Unmeasurable ^{d)}	-	-	-
9	0.384	Ir-F	60	Unmeasurable ^{d)}	-	-	-
10	0.768	Ir-F	12	74	2100	3500	1.66

Ir-F = (4,4'-Di-tert-butyl-2,2'-bipyridine)bis[3,5-difluoro-2-[5-(trifluoromethyl)-2-pyridinyl]phenyl]iridium(III) Hexafluorophosphate.

a) 9-mesityl-10-methylacridinium perchlorate. b) Visible light. c) Photocatalyst (10 mol.%). d) Gelatinous components, which are insoluble in any solvent, are formed. MeCN, acetonitrile.

Figure 3. Characteristics of the biomass-derived decomposable polymers. (A) Structures of **P1–P6**. (B) Thermogravimetric analysis (TGA) thermograms of **P1–P6**. (C) Differential scanning calorimetry (DSC) thermograms of **P1–P6**. (D) Thermal properties of each polymer. Polymerization conditions, **P1**: 0.096 mM **M1** in HFIP initiated by 100 mol.% PIDA at 25 °C for 60 min. **P2** and **P4–P6**: 0.384 M monomer in acetonitrile (MeCN) catalyzed by 1 mol.% of the Ir-F photocatalyst at 25 °C for 60 h. **P3**: 0.384 M **M3** in MeCN catalyzed by 1 mol.% of the Ir-F photocatalyst at 25 °C for 17 h.

Comment: 3. In the paragraph on page 14 about Chemical re- and upcycling, the lack of molecular weight makes it less reader friendly. It is easier to understand if the authors compare the molecular weight of the polymer after the initial polymerization with that of the recycled polymer.

Response: Thank you for this helpful suggestion. We agree that including molecular weight information for the recycled products improves the clarity and accessibility of the manuscript. In response, we have added the M_n and PDI values of the initially synthesized **P1**, the recycled **P1**, and the upcycled polyurethane to Figure 5 in the revised manuscript. Notably, the molecular weight of recycled **P1** was almost identical to that of the original **P1**, demonstrating that the material can be depolymerized and repolymerized without significant loss of structural integrity. This supports the feasibility and robustness of our chemical recycling strategy. As for the epoxy-cured resin, due to its cross-linked nature and insolubility in common solvents, GPC analysis and molecular weight determination was not feasible.

These updates ensure that all available characterization data for the upcycled materials are now explicitly reported.

Figure 5. Re- and upcycling of **B1**. (A) Repolymerization to yield **P1** by adding 1 equiv. of dichlorodiisopropylsilane. (B) Synthesis of polyurethane via reaction with 4,4'-methylenebis(phenyl isocyanate) in the presence of 1,4-diazabicyclo[2.2.2]octane. (C) Epoxidation of **B1** via reaction with epichlorohydrin. (D) Curing reactions of epoxide **E1** with curing agents, such as 4,4'-methylenebis(cyclohexylamine) and isophoronediamine, to yield thermoset networks.

The manuscript discusses a novel approach to producing decomposable polymers with high biomass contents *via* [2 + 2] cycloaddition polymerization using biomass-derived bifunctional monomers, phenyl propanoids with silyl ether linkages. This comprehensive work systematically investigates the polymerization reaction using chemical, electrochemical, and photochemical methods employing a well-defined library of monomers. Although the molecular design and polymerization methods are innovative, the obtained molecular weights of polymers were modest, especially with chemical and electrochemical processes; at the same time, most of the monomers' reactivity was limited. The resulting polymers contained aromatic, cyclobutane rings and silyl ether bonds in their backbones, providing thermal stability and functionality for selective decomposition via Diels-Alder reactions at the cyclobutene rings and fluoride ion addition at Si-O bond cleavage, also providing pathways for the recycling and upcycling. This study comprehensively investigates polymerization, selective depolymerization, and re- and upcycling of high-biomass content decomposable polymers, which could be valuable in achieving a circular economy. The manuscript can be recommended for publication after addressing the following comments:

1. In the case of hole-catalytic polymerization using a chemical oxidant, PIDA, the reaction conversion reached 62% of M1 within 1 minute using 10 mol% of PIDA. This indicates that the reaction rate is swift, enhancing the possibility of a competing termination reaction during chain propagation. Considering such a low molecular weight of P1 (~2300 g/mol), is it possible to investigate the addition of lower equivalents or even sequential aliquots of PIDA?
2. The monomers M2-M6 did not show polymerization in the case of chemical oxidation and electrochemical oxidation. The authors justified this behavior with higher oxidation potential as observed in cyclic voltammograms (CV) in Figure S2; however, a better justification would be needed if 5 of the 6 synthesized monomers do not yield polymerization. The oxidation potential should be identified from CV to calculate the HOMO/LUMO levels, along with DFT calculations, if available.
3. In contrast, M1 could not be polymerized with photocatalysis, which is an ambiguous behavior. This phenomenon requires more explanation.
4. Interestingly, polymer metabolites B1-B6 obtained from Si-O bond breakage showed regioselectivity at methine protons of the cyclobutane rings, favoring head-to-head over head-to-tail. Since these monomers were obtained only from Si-O bond breakage, similar regioselectivity at methine groups could be identified in the polymers, especially when the molecular weights are moderate.
5. Polyurethane, epoxy-cured resin, and even P1 via polycondensation were synthesized during the demonstration of upcycling, which is very promising. The polymer characterization data, such as molecular weights and PDI (GPC traces), should be included.
6. The citation of Figure 1D is missing in the main text; the First citation of Figure S3 is wrongly correlated with GPC traces.

Reviewer Comments for Nature Communications (NCOMMS-25-27860)

Dear Editor and Authors

In this article, the authors showed a sustainable method for the synthesis of high-biomass decomposable polymers via [2+2] cycloaddition polymerization. The polymers obtained exhibited excellent thermal properties and underwent efficient dual decomposition via Diels-Alder reactions and Si-O bond cleavage. The schemes and figures were easy to understand, there were no careless mistakes, and the paper felt very easy to read. The concept of this research topic is intriguing. However, there are still several hurdles to overcome before it can be published. The revision is needed before acceptance for publishing in “*Nature Communications*”.

1) At the end of the second paragraph of their three-page introduction, the authors write:

Moreover, to the best of our knowledge, no studies regarding decomposable polymers prepared using phenylpropanoids, which are desirable for the sustainability of the polymer industry, have been reported (Figure 1A).

However, a paper with a similar concept was published in 2024 with the following title:

Synthesis of photodegradable polyesters from bio-based 3,4-dimethoxycinnamic acid and investigation of their degradation behaviors (DOI: 10.1016/j.polymer.2024.127204)

As this paper was found through a quick search, the authors should conduct a more careful literature search and cite papers for comparison where necessary.

- 2) The molecular weights of the polymers synthesized this time are generally low, with some having M_w below 4000. It seems difficult to compare the thermal properties of TGA and DSC at this molecular weight and say that it is the effect of a particular functional group.
- 3) In the paragraph on page 14 about *Chemical re- and upcycling*, the lack of molecular weight makes it less reader friendly. It is easier to understand if the authors compare the molecular weight of the polymer after the initial polymerization with that of the recycled polymer.

Sincerely,